# Importance of Tyrosine Phosphorylation in Hormone-Regulated Plant Growth and Development

**DOI:** 10.3390/ijms23126603

**Published:** 2022-06-13

**Authors:** Weimeng Song, Li Hu, Zhihui Ma, Lei Yang, Jianming Li

**Affiliations:** 1State Key Laboratory for Conservation and Utilization of Subtropical Agro-Bioresources, College of Forestry and Landscape Architecture, South China Agricultural University, Guangzhou 510642, China; wmsong@scau.edu.cn (W.S.); hl@stu.scau.edu.cn (L.H.); mzh@stu.scau.edu.cn (Z.M.); 20213154081@stu.scau.edu.cn (L.Y.); 2Guangdong Key Laboratory for Innovative Development and Utilization of Forest Plant Germplasm, College of Forestry and Landscape Architecture, South China Agricultural University, Guangzhou 510642, China; 3Department of Molecular, Cellular, and Developmental Biology, University of Michigan, Ann Arbor, MI 48109, USA

**Keywords:** protein tyrosine phosphorylation, phytohormones, plant growth and development, brassinosteroids

## Abstract

Protein phosphorylation is the most frequent post-translational modification (PTM) that plays important regulatory roles in a wide range of biological processes. Phosphorylation mainly occurs on serine (Ser), threonine (Thr), and tyrosine (Tyr) residues, with the phosphorylated Tyr sites accounting for ~1–2% of all phosphorylated residues. Tyr phosphorylation was initially believed to be less common in plants compared to animals; however, recent investigation indicates otherwise. Although they lack typical protein Tyr kinases, plants possess many dual-specificity protein kinases that were implicated in diverse cellular processes by phosphorylating Ser, Thr, and Tyr residues. Analyses of sequenced plant genomes also identified protein Tyr phosphatases and dual-specificity protein phosphatases. Recent studies have revealed important regulatory roles of Tyr phosphorylation in many different aspects of plant growth and development and plant interactions with the environment. This short review summarizes studies that implicated the Tyr phosphorylation in biosynthesis and signaling of plant hormones.

## 1. Introduction

Protein phosphorylation, which was first reported almost 50 years ago [1], is a crucial post-translational modification (PTM) that occurs on more than 70% of all known proteins in eukaryotes [2]. Protein phosphorylation plays important roles in numerous signaling pathways via regulating the activities, localizations, stabilities, and biochemical interactions of proteins [3,4,5,6]. Serine (Ser), threonine (Thr), and tyrosine (Tyr) are the three residues that are often phosphorylated on proteins in eukaryotes, with the ratios of ~86:12:2% [7]. Although Tyr phosphorylation only accounts for less than 2% of all known phosphorylated residues, it was implicated in a wide range of signaling processes of eukaryote physiology, especially in animals. The dynamic of protein phosphorylation level in cells is controlled through the combined action of protein kinases and phosphatases. Many different Tyr kinases regulating diverse physiological processes were reported in animals, but plant genomes encode no Tyr-specific kinases [8,9]. However, recent studies have demonstrated that the Tyr phosphorylation is equally prevalent in plants as in animals [10,11]. It was shown that the detected Tyr phosphorylation in plants is attributed to the abilities of some plant protein Ser/Thr kinases to phosphorylate not only Ser and Thr residues but also Tyr residues. Given the importance of these characterized plant dual-specificity kinases in all aspects of plant growth and development, it was suggested that Tyr phosphorylation must be a crucial regulatory mechanism in plants.

The Tyr phosphorylation has been implicated in many plants’ growth and development and is regulated by several plant hormones, especially the plant steroid hormones brassinosteroids (BRs). Mass spectrometry (MS)-assisted identification of phosphorylation sites detected the presence of phosphorylated Tyr residues on the surface-localized receptor complexes and intracellular BR signaling components [12,13,14,15]. Recent studies have also suggested an important regulatory function of Tyr phosphorylation in the signaling processes of other plant hormones [16,17,18]. Furthermore, Tyr phosphorylation was also found to play crucial roles in plant stress tolerance, including the pathogens’ induced immunity and resistance to abiotic stresses such as drought and salinity [9,19,20]. However, our short review only covers the studies that demonstrated important regulatory functions of the Tyr phosphorylation in the signaling pathways of phytohormones-mediated plant growth and development, with a general discussion on protein kinases and phosphatases that are either specific to Tyr residue or exhibit dual specificity toward Ser/Thr and Tyr residues.

## 2. PTKs and Dual-Specificity PTKs

Protein tyrosine kinases (PTKs) is a class of protein kinases that catalyze the transfer of the γ-phosphate group from ATP to the Tyr residues of substrate proteins. PTKs are well-studied in metazoan and are known to regulate cell proliferation signaling pathways. Depending on the kinase structure, PTKs can be divided into two categories: receptor PTKs (RTKs) and non-receptor PTKs (NRTKs). The human genome encodes a total of 90 PTKs, of which 58 are annotated as RTKs that can be further grouped into twenty subfamilies [21,22]. RTKs exhibit conserved molecular architectures, consisting of an extracellular ligand-binding domain, a single transmembrane helix, and a cytoplasmic portion that contains the Tyr kinase domain plus additional carboxy (C-) terminal and juxtamembrane regulatory regions [22,23]. NRTKs generally lack receptor-like features such as the extracellular ligand-binding domain and the transmembrane region. They usually present in the cytoplasm or are anchored to the cell membrane. In addition to the Tyr kinase domain, NRTKs usually possess domains known to mediate protein–protein, protein–lipid, and protein–DNA interactions [21,23]. Alteration in RTKs and aberrant activation of their intracellular signaling pathways can cause a series of human diseases, including cancers, diabetes, inflammation, severe bone disorders, and angiogenesis [22]. Given their important roles in cell proliferation, PTKs are largely associated with tumorigenesis, as well as tumor invasion and metastasis [21,24,25], thus making PTKs the hot targets for antitumor drug research.

To date, no evidence has been found for the existence of classical PTKs in plants [8,9]. However, the percentage of phosphorylated Tyr residues of all phosphorylated residues in plants was similar to that found in animals [10,11]. It was thought that this is attributed to the presence of dual-specificity protein tyrosine kinases (DsPTKs) that can phosphorylate not only Ser/Thr but also Tyr residues in plants. An early genomic survey using the conserved Tyr phosphorylation motifs in metazoan revealed the existence of 57 DsPTKs in Arabidopsis [8]. Plant DsPTKs and typical animal PTKs share a common kinase domain of around 250 amino acids containing 11 conserved subdomains. The sub-domain VI confers Ser/Thr specificity, while sub-domains VIII and XI confer Tyr kinase specificity [8]. All the identified 57 kinases possess both the Ser/Thr-specificity motifs in subdomains VI and the Tyr-specificity motifs in subdomain XI, suggesting that all of these kinases belong to a DsPTK family [8]. Recent studies have shown that many important plant receptor-like kinases (RLKs), sharing similar architectures with metazoan RTKs, exhibited the dual-specificity kinase activities, indicating that the Tyr phosphorylation is also an important regulatory mechanism in plants [19,26]. A recent in silico analysis in rice genome developed by Allimuthu et al. identified 18 DsPTKs which share >80% identity within the conserved sequences of subdomains VI, VIII, and XI with the Arabidopsis Brassinosteroid-Insensitive1 (BRI1) [27], a cell surface-localized leucine-rich repeat (LRR) RLK that functions as the BR receptor [28]. Tissue-specific and abiotic-stress-induced expression of these DsPTKs suggested their potential roles in regulating development and stress resistance in rice [27]. In addition, an earlier wheat cell-free system-based protein array study using 759 Arabidopsis protein kinases identified 38 kinases capable of autophosphorylation at Tyr residues, including members of Glycogen Synthase Kinase 3 (GSK3), RLK, casein kinase, Mitogen-Activated Protein Kinase (MAPK), and calcium-dependent protein kinase-related kinase (CRK) families [29]. Further investigation of these DsPTKs will greatly enhance our knowledge of the roles of Tyr phosphorylation in signaling processes of plant growth, development, stress resistance, and immunity.

## 3. PTP and Dual-Specificity Protein Phosphatases

The protein tyrosine phosphatases (PTPs) are enzymes functioning in a coordinated manner with PTKs to control the Tyr phosphorylation status of many protein substrates, thus regulating many signaling pathways involved in diverse physiological processes. Altered expression and/or activity of PTPs can lead to dysregulation of Tyr phosphorylation in cells and cause a wide range of human diseases such as tumorigenesis, autoimmune disorders, and abnormal skeletal development [30,31]. So far, at least 126 putative PTPs have been found in the human genome. In addition to the Tyr residue, some members of this large family exhibit other substrate specificities, such as Ser/Thr residues, inositides, glycogens, mRNAs, or even inorganic moieties [20,32,33,34]. Based on the nucleophilic catalytic residues in their catalytic motifs and topology, these PTPs can be broadly grouped into three families: cysteine (Cys)-, aspartic acid (Asp)-, and histidine (His)-based PTPs [30,34,35,36]. The Cys-based PTPs family, which is characterized by conserved signature motif (H/V)CX_5_R(S/T) (H, histidine; V, valine; R, arginine; and X for any amino acid residue) in the catalytic domain, is the largest group that can be divided further into three subgroups: classes I, II, and III PTPs based on their evolutionary lineages [30,34]. Class I consists of up to 117 PTPs, accounting for more than 95% of the Cys-based PTP family, which can be further divided into six subclasses: the classical phosphotyrosine (pTyr)-specific phosphatases, Vaccinia virus H1(VH1)-like/dual-specificity phosphatases (DUSP), SAC (suppressor of yeast actin mutations) phosphoinositide phosphatases, phosphatase domain-containing paladin 1 (PALD1), INPP4 (inositol polyphosphate 4-phosphatase) phosphatases, and TMEM55 (transmembrane protein 55A) phosphatases. Class II only contains two members: Low-Molecular-Weight PTPs (LMW-PTPs) and SSU72 (suppressor of SUa7, gene 2). Class III PTPs include three mammalian CDC25 proteins participating in cell-cycle regulation. Four eyes absent (EYA) phosphatases, previously categorized as Asp-based HAD (haloacid dehalogenase) phosphatases, are the only members of the Asp-based PTP family. The His-based PTPs include two ubiquitin-associated (UBA) and Src homology 3 (SH3) domain-containing protein (UBASH3) PGM (phosphoglycerate mutase) phosphatases, and three Acid Phosphatases (ACPs) [34].

Although PTPs are well characterized in mammalian cells, our knowledge on plant PTPs is rather limited. The first identified PTP in plants was AtPTP1, the only classical PTP encoded by the Arabidopsis genome [37]. Later on, different types of PTPs were identified from plants [38,39,40]. Compared with the human genome that encodes as many as 126 PTPs, the numbers of plant PTPs are much smaller. The Arabidopsis genome encodes a total of 47 putative PTPs, including 1 Tyr-specific PTP, 22 DUSPs, 1 LMW-PTP, and 23 Asp-based PTPs [38]. An earlier cell-free system that analyzed the in vitro dephosphorylation activity with 82 Arabidopsis protein phosphatases detected the Tyr-dephosphorylation activity for the PTP1 and three DUSPs [41]. A recent study found that the Arabidopsis genome encodes a novel Tyr-specific phosphatase, known as Rhizobiales/Rhodobacterales/Rhodospirillaceae-like Phosphatase 2 (RLPH2, due to its sequence similarity to bacterial protein phosphatases in the named three bacterial groups), which is highly conserved in the plant kingdom but lacks a homolog in animals [42]. It is important to note that, in contrast to protein Ser/Thr-specific, Tyr-specific, or dual-specificity kinases that share the same structural fold and similar catalytic mechanisms, different classes of PTPs exhibit different structural folds with quite distinct catalytic mechanisms. With increasing numbers of sequenced plant genomes and better bioinformatic tools and experimental toolkits, many PTPs with versatile physiological functions were identified in other plant species [40,43,44]. An analysis of the rice genome revealed the presence of 23 DUSPs and a single member of the classical PTP and LMWPTP in rice [40,43]. Searching homologs of annotated Arabidopsis/rice phosphatases identified a total of 159 protein phosphatases in maize, including 29 potential PTPs [44]. Although the number of Tyr-specific PTP is rather small in plants, the existence of many plant DUSPs suggested the physiological importance of dynamic regulations of the Tyr phosphorylation in plant growth and development.

## 4. The Role of Tyr Phosphorylation in Plant Growth Development

Plant growth and developmental processes are known to be orchestrated by different phytohormones, such as BR, gibberellins (GAs), auxin, cytokinin, ethylene, and abscisic acid (ABA). In this section, we discuss experimental evidence that support important roles of the Tyr phosphorylation in signaling/biosynthesis of these six plant hormones.

### 4.1. BR

BRs are a special class of plant hydroxysteroids that are essential for a broad range of plant growth and developmental processes [45,46,47,48,49]. Upon perception of BR, the receptor kinase BRI1 forms a complex with its co-receptor BRI1-Associated Kinase 1 (BAK1) [28,50,51,52] (Figure 1). Through heterodimerization and reciprocal phosphorylation, BRI1 and BAK1 are activated, resulting in the phosphorylation and activation of BR-Signaling Kinases 1(BSK1)/ Constitutive Differential Growth 1(CDG1) and bri1 Suppressor 1 (BSU1) [53,54].

Activated BSU1 was thought to inhibit, via Tyr dephosphorylation, the activity of a key negative regulator Brassinosteroid-Insensitive 2 (BIN2) [15], thus relieving its inhibitory effect on the activity of downstream transcription factors that regulates the expression of thousands of BR-responsive genes [55,56]. Clearly, protein phosphorylation is indispensable for the BR signaling pathway. It is worth noting that Tyr phosphorylation and dephosphorylation were recently implicated in regulating the intracellular signal transduction of the extracellular BR signals, from the kinase domain of the receptor BRI1 to downstream signaling components (Figure 1 and Table 1).

Classified as a canonical Ser/Thr RLK, BRI1 was surprisingly found to be autophosphorylated on Tyr residues in addition to Ser/Thr residues [12], indicating that BRI1 is a dual-specificity kinase. The dual-specificity nature of BRI1 is consistent with its crystal structures, since the BRI1 activation loop conformation resembles the Ser/Thr kinases, whereas the interaction of the phosphorylated Ser1044 in the activation loop with the phosphate-binding pocket consisting of three Arg residues (R922, R1008, and R1032) is reminiscent of typical Tyr kinases [57]. Although no in vivo phosphotyrosine (pTyr) residue of BRI1 was detected by liquid chromatography–tandem mass spectrometry (LC–MS/MS), three Tyr residues (Y831, Y956, and Y1072) of BRI1 (Table 1) were believed to be phosphorylated through analysis, using site-directed mutagenesis and phosphorylation/site-specific antibodies [12]. While pY831 in the juxtamembrane domain had a little impact on the overall kinase activity of BRI1, it reduces the in vivo BRI1 signaling activity, because expression of the mutant *BRI1^Y831F^-Flag* only partially rescued the dwarf phenotype of a weak BRI1 mutant, *bri1-5* [12]. Earlier studies also showed the essentiality of the two other Tyr residues (Y956 and Y1072) for the kinase activity of BRI1, since their Tyr-Phe mutations abolished the BRI1 kinase activity in vivo and in vitro [12]. However, it remains to be tested whether the reduced ability of the mutant BRI1^Y831F^ receptor to rescue the *bri1-5* mutation or abolished kinase activities of BRI1^Y956F^ and BRI1^Y1072F^ are truly caused by loss of the Tyr phosphorylation or caused by the elimination of the hydroxyl group (via the Tyr-Phe mutation) that could be important for the correct 3D structure of the BRI1 kinase or interacting with its coreceptor BAK1.

BAK1 functions as a coreceptor of BRI1 to help activate the BRI1’s kinase activity and to initiate BR signaling at the cell surface [51,52]. Similar to BRI1, BAK1 was also suggested to be a dual-specificity kinase, although no phosphorylated Tyr residue has been detected on endogenous BAK1 by mass spectrometry [13,58,59,60]. BAK1 was previously reported to be autophosphorylated on Y610 (Table 1) in the C-terminal domain in vitro and in vivo, using site-directed mutagenesis and phosphorylation-/site-specific antibodies [61]. However, the physiological function of Y610 phosphorylation on BAK1 remains unclear. It was previously thought that pY610 plays an important role in BR signaling and innate immunity [61], but a later study from the same laboratory concluded that the Y610 phosphorylation had no impact on either BR signaling or plant defense [62]. Y403 (Table 1) is another potential Tyr residue with its phosphorylation status affecting the in vivo functions of BAK1, which also functions as the essential coreceptor for many other ligand-binding RLKs including several plant innate immunity receptors [13,63,64]. It was shown that a loss-of-phosphorylation mutation of Y403 to Phe had little impact on BR signaling but abrogated its function in plant innate immunity [13].

In the absence of BRs, BRI1 is kept in its inactivated state by its inhibitory C-terminal tail and its binding to BRI1 Kinase Inhibitor 1 (BKI1) [45,68]. It was thought that BKI1 interacts with BRI1 at the plasma membrane to prevent the BRI1–BAK1 dimerization, thus keeping BRI1 at its inactive state. The BR-triggered BRI1 activation leads to BKI1 phosphorylation, resulting in the rapid dissociation of BKI1 from the plasma membrane into the cytosol [68]. Interestingly, BKI1 was found be phosphorylated at Y211 (Table 1) by BRI1 within a [K/R] [K/R] membrane binding motif in a BR-dependent manner, leading to the release of BKI1 into the cytosol and a full activation of BR signaling [14]. Thus, Tyr phosphorylation plays an essential role in initiating the BR signaling at the plasma membrane.

BIN2, one of the ten GSK3-like kinases in Arabidopsis, negatively regulates the BR signaling by phosphorylating downstream transcription factors, including bri1-EMS Suppressor1 (BES1) and Brassinazole-Resistant1 1(BZR1) [55,69,70]. A phosphorylated Tyr residue (pTyr200) (Table 1) on BIN2 is widely believed to be required for the kinase activity, since overexpression *BIN2^Y200F^–YFP* in Arabidopsis failed to cause the severe dwarf phenotypes that were often observed in *BIN2–YFP* overexpressing lines [15]. However, the involvement of pTyr200 in BIN2 regulation of BIN2 needs further confirmation, as the Y200F mutation could simply be explained by the substitution of a polar residue (carrying the hydroxyl group by a nonpolar residue and the consequential structural alteration [45]. It is important to note that it is also controversial about the involvement of the phosphorylation of the corresponding Tyr residue in the regulation of animal GSK3 [71,72,73,74,75]. Previous studies have shown that it is BSU1, a member of a plant-specific family of protein phosphatases with Kelch-like domain (PPKLs), that dephosphorylates the pTyr200 of BIN2 to active the BR response [15]. However, BSU1 is a canonical protein Ser/Thr phosphatase (PSP) and possesses a conserved PSP structure that is quite different from the conserved structural features of the catalytic domain of PTPs/DUSPs carrying the signature catalytic CX_5_R motif [45], which could prevent BUS1 to bind a pTyr residue. A previously published study revealed that BSU1 is only expressed in pollens and only presents in the Brassicaceae family with large sequence variations among BSU1 orthologs. Importantly, although overexpression of *BSU1* or its homologs could amplify BR signaling, eliminating BSU1-Like 2 (BSL2) and BSL3, which are close homologs of BSU1 and are known to be expressed in vegetative tissues, only resulted in a marginal impact on BR signaling, measured by BR-induced change in the phosphorylation status of BES1/BZR1, the two best-studied BIN2 substrates. Thus, it remains to be fully investigated to determine if the BSU1/BSLs-mediated pTyr200-dephosphorylation is truly involved in the BR-induced BIN2 inhibition to allow the extracellular BR signal to reach the nucleus of the BR-responsive cells [76].

### 4.2. GA

Gibberellins (GAs) are another class of plant hormones that also regulate many developmental processes in plants, including seed germination, stem elongation, leaf expansion, trichome development, pollen maturation, and the induction of flowering [77,78]. Upon binding GA, the GA receptor Gibberellin Insensitive Dwarf 1 (GID1) forms a complex with DELLA, a key negative regulator of GA signaling named after its five conserved amino acids (aspartate–glutamate–leucine–leucine–alanine), leading to ubiquitination and subsequent degradation of DELLA by the 26S proteasome and activation of downstream GA signaling events in the nucleus [78,79]. Previous studies have demonstrated that the GA-triggered DELLA degradation was regulated by phosphorylation at Ser/Thr, but not Tyr residues of DELLA [80,81,82]. However, the GA-induced DELLA degradation in barley and tobacco BY-2 cells could be inhibited by treatment with a broad Tyr kinase inhibitor genistein [83,84], suggesting the involvement of the Tyr phosphorylation in GA-induced DELLA degradation in certain plant species [16]. Consistent with this hypothesis, a recent investigation revealed a role of Tyr phosphorylation in ubiquitin-dependent degradation of GA receptor GID1 [16]. It was shown that GA Receptor RING E3 Ubiquitin Ligase (GARU), an E3 ubiquitin ligase responsible for degrading GID1, was phosphorylated at Tyr321 by Target of Genistein Kinase 2 (TAGK2) (Table 1), which is a member of the angiosperm-specific family CRK known to exhibit dual-specific kinase activity [29]. The phosphorylation of GARU at Tyr321 inhibited the GARU–GID1 interaction, thus stabilizing GID1, promoting the GA-dependent degradation of DELLA, and activating GA signaling [16] (Figure 2).

### 4.3. Auxin

Auxin, the first plant hormone discovered in the 1930s, regulates all aspects of plant growth and developmental processes throughout the entire plant life cycle [85,86]. It has been well established that the expression of many auxin-responsive genes is activated by Auxin Response Factors (ARFs) that form inactive transcriptional complexes with the Auxin/Indole-3-Acetic Acid (Aux/IAA) transcriptional repressors. Upon auxin binding, the auxin receptors, Transport Inhibitor Response 1/Auxin Signaling F-Box proteins (TIR1/AFBs), bind and ubiquitinate Aux/IAA, leading to proteasome-mediated degradation of Aux/IAA, activation of ARFs, and expression of many auxin-responsive genes [87]. Although the auxin signaling mechanism mainly involves protein degradation and transcriptional regulation, earlier studies did implicate a role of protein Tyr phosphorylation in auxin signaling. It is well established that Indole-3-butyric Acid Response 5 (IBR5), an Arabidopsis DUSP, is an important positive regulator of auxin response [17]. Moreover, *ibr5* mutants, which are less sensitive to synthetic auxins and auxin transport inhibitors, exhibited similar growth phenotypes with other auxin signaling mutants [17,18]. Importantly, a double mutant of *ibr5-1* and an auxin receptor mutant, *tir1*, exhibited enhanced auxin-response defects, suggesting that IBR5 likely regulates auxin signaling independently of the TIR1 auxin receptor [17]. To date, the only substrate for IBR5 is Mitogen Activated Protein Kinase 12 (MPK12), a MAP kinase that was known to negatively regulate the auxin pathway [88]. IBR5 interacts and dephosphorylates MPK12, and RNAi-mediated suppression of *MPK12* in *ibr5* partially rescues the auxin-insensitivity phenotype [88]. Given the fact that Tyr phosphorylation of MPKs is required for their activation [89,90], these studies support a potential role of Tyr phosphorylation in modulating the auxin signaling. A recent study also implicated MPK6 and MAP Kinase Phosphatase 1 (MKP1), a known DUSP that could dephosphorylate pSer/pThr/pTyr residues [91], in a growth response of Arabidopsis roots to L-Glutamate by regulating auxin distribution and response [92]. A recent chemical biology study has demonstrated a role of the auxin-induced phosphorylation at Tyr248 residue (Figure 2 and Table 1) of an Arabidopsis Receptor for Activated C Kinase 1 (RACK1, a WD-40 type scaffold protein) in the auxin-mediated lateral root development [65]. It was shown that a compound that stabilized the Tyr248 phosphorylation stimulated lateral root formation, whereas two other small compounds that inhibited the auxin-induced Tyr248 phosphorylation reduced auxin sensitivity and interfered with the lateral root development. It would be interesting to know if the auxin-induced RACK1 Tyr248 phosphorylation involves a balancing act of a MPK and a DUSP. In addition to MPKs, Senescence-Associated Receptor-Like Kinase (SARK), a soybean LRR-RLK with dual-specificity kinase activity, was found to influence the auxin and ethylene pathways [93]. It was reported that overexpression of the *GmSARK* in Arabidopsis promoted both synthesis and response of auxin and ethylene, whereas a mutation of either *Auxin Resistant 1* (*AUX1* encoding an auxin influx carrier) or *Ethylene Insensitive 2* (*EIN2* encoding an ER membrane-localized positive ethylene signaling component) completely rescued the senescence phenotype caused by *GmSARK* overexpression [93]. Further studies are needed to fully understand the biochemical mechanisms by which Tyr phosphorylation regulates the biosynthesis, transport, or signaling of auxin.

### 4.4. Cytokinin

The phytohormone cytokinin is also indispensable for plant development, regulating many development and physiological processes such as meristem development, nutrient response, senescence, and stresses resistance [94,95]. The cytokinin signal transduction pathway involves a phosphorelay between histidine (His) and aspartate (Asp) residues among three main signaling components. The first components are “hybrid” histidine kinase (HK) receptors containing the cytokinin-binding CHASE (Cyclases/Histidine Kinases Associated Sensory Extracellular) domain, histidine kinase, and receiver domain, while the second components are histidine phosphotransferases (HPs) that shuttle phosphoryl groups between cytosol and nucleus. The third signaling components are two different types of Response Regulators (RRs), which function either as DNA-binding transcription factors mediating cytokinin-regulated genes expression (B-type RRs) or as negative regulators of cytokinin signaling (A-type RRs) via their inhibitory actions on B-type RRs [94,95,96]. Despite using the His-Asp phosphorelay as the main signaling mechanism, recent studies have demonstrated the involvements of Tyr phosphorylation in regulating cytokinin biosynthesis. In rice, the transcription factor drought and salt tolerance (DST) directly induced the expression of *Cytokinin Oxidase2* (*OsCKX2*), which encodes a cytokinin oxidase catalyzing the degradation of active cytokinins [97]. It was shown that the rice OsMKKK10–OsMKK4–OsMPK6 cascade positively regulates the *DST* transcription, thus suppressing the cytokinin content in rice [98,99] (Figure 2). Interestingly, a comparative phosphoproteomics analysis of wild type and the *ahk2ahk3* double mutant, which lacks two cytokinin receptors AHK2 (Arabidopsis Histidine Kinase 2) and AHK3, revealed a Tyr213 phosphorylation on CKX2 in Arabidopsis (Table 1), which may modulate its enzymatic activity [66]. The same dataset also identified an enhanced Tyr19 phosphorylation of KIN10 (Sucrose Non-Fermenting 1 (SNF1)-Kinase homolog 10) [66]. Given that KIN10 is a catalytic subunit of the SNF1-Related Kinase 1 (SnRK1) that acts as an energy sensor, this finding provided a link between cytokinin signaling and energy homeostasis in plants. Moreover, the MAP Kinase 18 (MPK18) was found to be less phosphorylated in the *ahk2ahk3* mutant compared with the corresponding wild-type plants, suggesting a potential connection between the His–Asp phosphorelay mechanism with the conserved MAPKKK–MAPKK–MAPK signaling cascade, which is well-known to involve Thr/Tyr phosphorylation in regulating the MAPK activity [89,90], in the cytokinin signaling [66]. Consistently, another study also showed that both gene expression and kinase activities of OsMPK3 and OsMPK6 were responsive to exogenous cytokinin treatment [100]. Moreover, cytokinin is required for the Tyr dephosphorylation and activation of Cyclin-Dependent Protein Kinase (CDK), which is necessary for cell division in cultured *Nicotiana plumbaginifolia* cells [101]. Together, these recent studies have provided strong evidence for an important role of Tyr phosphorylation in regulating the cytokinin biosynthesis and signaling.

### 4.5. Ethylene

Similar to cytokinin, the signaling mechanism of ethylene, the first gaseous hormone ever discovered in any living organism, is also initiated by histidine-like kinases that include Ethylene Response 1 (ETR1) [102]. Upon ethylene binding, ETR1 and its homologous ethylene receptors undergo conformational changes, resulting in the inhibition of a kinase known as Constitutive Triple Response 1 (CTR1), which negatively regulates ethylene signaling through its direct phosphorylation (at Ser645 and Ser924) of the ER membrane-anchored Ethylene-Insensitive 2 (EIN2) [103]. The inhibition of CTR1 permits cleavage of non-phosphorylated EIN2, allowing nuclear translocation of the cleaved C-terminal EIN2 domain and subsequent nuclear accumulation of downstream transcription factors EIN3/EIL1(EIN3-Like1) and Ethylene Response Factors (ERFs) that regulate expression of many ethylene responsive genes [102,103,104]. CTR1 is a protein kinase with sequence homology to the mammalian Raf-like MAPKKK kinases [105], suggesting that the ethylene signaling mechanism might involve a MAPKKK–MAPKK–MAPK cascade with MAPKK being the dual-specificity kinase. Early studies seemed to support this hypothesis. It was reported that two Medicago MPKs (SIMK for stress-induced MAPKK and MMK3 for Medicago MAP kinase 3) and the Arabidopsis MPK6 could be activated by aminocyclopropane-1-carboxylic acid (ACC) that is the immediate biosynthetic precursor of ethylene. Interestingly, the ACC-triggered activation of MPK6 was dependent on ETR1 and CTR1 but independent on EIN2 and EIN3, suggesting a potential placement of a MAPK module between CTR1 and EIN2 [106]. Another Arabidopsis study suggested the involvement of a MKK9–MPK3/MPK6 module in the ethylene signal transduction pathway [107]. Ectopic expression of an active *MKK9* in Arabidopsis confers a constitutive ethylene phenotype in etiolated seedlings, which could be suppressed by *ein2* but not *etr1-1* mutations, again placing this MKK9–MPK3/MPK6 cascade between CTR1 and EIN2 [107]. Intriguingly, MKK9-activated MPK6 could stimulate EIN3 phosphorylation at Thr174 residue to stabilize the nuclear EIN3 protein [107]. However, the involvement of such MAPK cascades in ethylene signaling remains controversial [108,109]. Debate mainly came from inconsistent findings of the MKK9 roles from different groups [110,111]. Moreover, it was shown previously that the MKK9–MPK3/6 cascade was actually involved in ethylene biosynthesis. In Arabidopsis, MPK3 and MPK6 phosphorylated and thus enhanced the stabilization of ACC Synthase 2 (ACS2) and ACS6, two important enzymes in the ethylene biosynthetic pathway, resulting in increased ethylene biosynthesis [112,113] (Figure 2). As discussed above, the MAPK cascade requires Tyr phosphorylation for activating MAPKs by the dual-specificity kinase MAPKK [89,90]; thus, unequivocal confirmation of its involvement in ethylene biosynthesis or signaling would definitely support an important role of the Tyr phosphorylation in plant ethylene responses.

Additional support for the involvement of protein Tyr phosphorylation in the plant ethylene response came from phosphoproteomic studies. A phosphoproteomic profiling experiment with ethylene/air-treated petunia corollas identified a total of 1443 ethylene-regulated phosphorylation sites, including 20 pTyr residues [114], while a phosphoproteomic study using several Arabidopsis ethylene-response mutants identified a total of 1067 pSer/pThr residues but only 22 pTyr sites [115]. Consistent with the two phosphoproteomic studies, an in vitro kinase assay discovered that ERF13, an ethylene response factor, could be phosphorylated at Tyr16/Tyr207 residues by the Arabidopsis dual-specificity kinase CRK3 [29] (Figure 2 and Table 1).

### 4.6. ABA

Abscisic acid (ABA) serves as a growth inhibitory plant hormone with important roles in seed maturation, seed dormancy, and stress tolerance in plants [116,117]. Upon perception of ABA, ABA receptors, Pyrabactin Resistant 1 (PYR1), and PYR1-Like proteins (PYLs) interact with several members of the type 2C Protein Phosphatases (PP2Cs), thus relieving their inhibitory effects on members of the SnRK2 subfamily [116,118,119,120]. The activated SnRK2s phosphorylate the downstream signaling components, including membrane-localized anion channels and nuclear transcription factors, leading to the activation of conserved ABA responses [121,122]. Although protein phosphorylation is crucial to ABA signaling, the role of Tyr phosphorylation was rarely reported. However, several studies using phenylarsine oxide (PAO), a potent PTP inhibitor, showed that PAO affected several ABA-dependent responses in plants, including the transcriptional induction of *RAB16* (Response to ABA 16) in barley protoplasts [123], ABA-dependent stomatal closure in *Commelina communis* [124], and seed germination in Arabidopsis [125]. Additional pharmacological studies demonstrated that inhibitors of PTKs, such as genistein, tyrphostin A23, and erbstatin, could also influence the ABA responses in Arabidopsis [126]. Together, these results strongly suggested the involvements of PTPs and PTKs in regulating ABA responses with no knowledge on whether these chemical inhibitors had direct impact on key biosynthetic enzymes or major signaling components of ABA. A recent study showed that the Arabidopsis Open Stomata 1 (OST1)/SnRK2.6, a critical component of ABA signaling, was a dual-specificity kinase capable of autophosphorylation at conserved Tyr/Ser residues (Tyr163, Ser164, Ser166, and Ser167) (Table 1) in its activation loop that are required for the full activation of OST1 [67]. Importantly, ABA treatment could induce Tyr182 phosphorylation, which could be dephosphorylated by ABA-Insensitive 1 (ABI1) and its homologs [67] (Figure 2). It is important to note that PP2Cs generally function as canonical PSPs and an earlier wheat cell-free system-based in vitro assay with 82 Arabidopsis protein phosphatases detected no pTyr-dephosphorylation activity for 49 of the Arabidopsis PP2C family of 69 members [41]. Further experiments are thus needed to verify whether the pTyr182 can be truly dephosphorylated by members of the Arabidopsis PP2C family.

Additional support for the involvement of Tyr phosphorylation in ABA signaling came from studies of several DUSPs. It was reported that several Arabidopsis loss-of-function mutants of Propyzamide-Hypersensitive 1 (PHS1) were hypersensitive to ABA during seed germination [127], whereas a loss-of-function mutant of PTP-like Nucleotidase (PTPN) was hyposensitivity to ABA to inhibit seed germination [128] (Figure 2). Arabidopsis DSPTP1 is an active DUSP that acts as a negative regulator of osmotic stress, as the *dsptp1* mutant shows enhanced tolerance to osmotic stress during seed germination and seedling establishment [129]. Moreover, elimination of the *DSPTP1* gene decreased the ABA content and suppressed the ABA-signaling pathway under osmotic stress, which could be attributed to altered expression of many genes associated with ABA biosynthesis and catabolism such as *NCED3* (9-*cis*-epoxycarotenoid dioxygenase 3) and *CYP707A4* [129] (Figure 2). Similarly, a rice DUSP, Plant and Fungi Atypical Dual-Specificity Phosphatase 1 (OsPFA-DSP1), could also negatively regulates the drought tolerance, because its overexpression in tobacco plants increased sensitivity to drought stress and inhibited the ABA-induced stomatal closure [130] (Figure 2). Together, these genetic and transgenic studies provided additional support for the involvement of Tyr phosphorylation in ABA-regulated processes. Further studies are needed to identify potential substrates of those DUSPs to have a better understanding on how the Tyr phosphorylation influences many ABA-mediated physiological processes.

### 4.7. Other Plant Growth Processes

In addition to functioning in hormone-mediated cellular processes, recent studies have shown that Tyr phosphorylation is also involved in many other aspects of plant growth and development. In carrots, protein Tyr phosphorylation patterns vary among different tissues or at different stages of somatic embryogenesis, and somatic embryogenesis could be blocked by treatment of a PTK inhibitor, tyrphostin A25 [131]. Furthermore, PTK inhibitors such as genistein prevented the establishment of zygotic polarity in *Fucus*, also suggesting the involvement of Tyr phosphorylation in embryo development [132]. Pollen development could also be regulated by Tyr phosphorylation. Functioning as a tumor suppressor, Phosphatase and Tensin homolog (PTEN) is a unique DUSP in animals, which dephosphorylates both pTyr residue and phosphatidylinositols [133]. AtPTEN1, an Arabidopsis homolog of PTEN, is expressed exclusively in pollen grains and is indispensable for pollen development [134]. Silencing of *AtPTEN1* by RNA interference causes pollen cell death during the late stages of pollen development [134]. Moreover, accumulation of autophagic bodies in pollen tubes was observed in *PTEN* overexpression transgenic lines, which was dependent upon its lipid phosphatase activity toward phosphatidylinositol 3-phosphate (PI3P), as exogenous application of PI3P or expression of a class III phosphatidylinositol 3-kinase (PI3K) that produces PI3P can rescue such effect [135]. Moreover, many *DsPTKs* were found to be specially expressed in stamen [8], further supporting involvement of Tyr phosphorylation in pollen development. In addition to PI3P, whether Tyr phosphorylated proteins that are implicated in pollen development are also substrates of PTEN would be an interesting question.

Tyr phosphorylation was also detected and/or implicated in regulating the dynamic organization of cytoskeleton proteins such as actins/profilins and microtubules. An earlier study reported a correlation between a reduction of actin Tyr phosphorylation in the pulvinus with touch-induced petiole bending in the touch-sensitive shameplant *Mimosa pudica* [136]. Tyr phosphorylation of profilins in the common bean (*Phaseolus vulgaris*) plants that binds actins, polyprolines, and inositol phospholipids was thought to generate binding specificities that likely regulate the dynamic balance between membrane trafficking and actin organization [137,138]. Treatment of Arabidopsis seedlings with specific inhibitors of RTKs and PTPs implicated a role of Tyr phosphorylation in regulating the dynamic organization of cortical microtubes, thus affecting root hair development and root growth [139]. Consistently, tubulins were found to be phosphorylated in Arabidopsis seedlings and cultured tobacco cells (*Nicotiana tabacum*) [140,141]. It was hypothesized that DsPTKs such as CRKs and Dual-Specificity Tyrosine Phosphorylation-Related Kinases (DYRKs) could be the kinases that phosphorylate the plant tubulins [29,142]. A recent genome-editing study revealed a crucial role of a DYRK in regulating cell shape and tissue morphogenesis in the lower plant liverwort (*Marchantia polymorpha*) [143], but whether this DYRK is involved in phosphorylating Tyr residues of cytoskeleton proteins requires further studies.

A review on the role of Tyr phosphorylation in plant growth and development is not completed without a discussion on photomorphogenesis, a series of photoreceptor-mediated morphological changes that allow plants to maximize their growth potential [144]. It was previously reported that the phosphorylation of the Tyr104 residue serves as an important regulatory mechanism to control the red-light photoreceptor phytochrome B (phyB) [145]. A phosphomimic mutant phyB^Y104E^ failed to rescue an Arabidopsis *phyb* loss-of-function mutant, whereas a loss-of-phosphorylation phyB^Y104F^ mutant photoreceptor conferred light hypersensitivity. Consistently, another study implicated the Arabidopsis Yet Another Kinase 1 (YAK1), a member of the highly conserved DYRK family [146], in several light-regulated responses, including photomorphogenesis, fertility, and flowering [147]. It would be interesting to test whether YAK1 is the kinase that phosphorylates the Tyr104 residue of phyB. It is important to note that recent studies also implicated YAK1 in regulating the meristem activity and ABA signaling in Arabidopsis via its interaction with the Arabidopsis Target of Rapamycin (TOR) kinase complex [148,149], a central regulator that integrates multiple developmental and environmental signals to regulate plant growth and stress responses [150].

## 5. Perspective

With the successful identification of many Tyr phosphorylated proteins in plants and demonstration of in vitro Ser/Thr-Tyr dual-specificity of many plant DsPTKs and PTPs/ DUSPs, it became clear that Tyr phosphorylation is a common regulatory mechanism involved in a wide range of plant growth/developmental processes and plant stress response pathways. However, compared to what is known in animal systems, our mechanistic understanding of the plant Tyr phosphorylation in regulating plant growth and development remains limited. A simple search at pubmed.gov with “tyrosine phosphorylation” returned 60,598 results, whereas a “plant tyrosine phosphorylation” query only generated 1,107 results. On the one hand, many phosphoproteomic studies detected many Tyr-phosphorylated proteins, but little is known about potential DsPTKs or PTPs/DUSPs involved in regulating their phosphorylation status in plants [11]. On the other hand, some of the presumed phosphorylated Tyr residues lacked support from MS-based verifications but relied on using pTyr-specific antibodies, which could be influenced by post-translational modifications of nearby residues or cross-react with other phosphorylated residues [151], and transgenic experiments with mutant transgene constructs carrying loss-of-phosphorylation Tyr-Phe or phosphomimic Tyr-Glu mutation that could alter the 3D structures of the corresponding mutant proteins [45]. Therefore, even though great progress was achieved, challenges still remain, including but not limited to the following:

(1) Although Tyr phosphorylation and dephosphorylation are essential for BR signaling, no PTPs/DUSPs that could dephosphorylate the components in BR signaling have been found, except for BSU1, a canonical PSP with a controversial function on pTyr200 of BIN2 [15]. The identification of true PTPs/ DUSPs regulating the BR signaling pathway will greatly enhance our understanding of how BR regulates plant growth and development.

(2) In vivo and mechanistic investigations are urgently needed. For most DsPTKs and PTPs/DUSPs, their functions are simply analyzed through physiological experiments with loss-of-function mutants. Detailed investigations such as identifying their substrates and corresponding pTyr residues are required to fully understand the underlying regulatory mechanisms of the observed developmental and physiological phenotypes. Even for these DsPTKs and PTPs/DUSPs with known substrates, most of the supporting evidence came from in vitro analysis but clearly lacked in vivo confirmation.

(3) Are “reader” proteins involved in the pTyr-based signal transduction processes in mammalian systems also present in plant cells? The “writer” and “eraser” functions are fulfilled by DsPTKs and PTPs/DUSPs in plant cells, respectively. It was well-known that the “reader” proteins in mammals can specifically recognize and bind certain pTyr residues on target proteins to propagate the phosphorylation in a wide range of signaling pathways [19]. It would be a challenging task to find plant-specific “reader” proteins or “reader” motifs important to regulate many phosphorylation-mediated signaling processes in plants.

Given the important roles of Tyr phosphorylation in plant growth and stress tolerance, additional experimental approaches are needed to gain further insights into how plants employ the Tyr phosphorylation to regulate a wide range of plant growth and developmental processes under varying environmental conditions. Improving the detection sensitivity and accuracy of MS-based quantification of in vivo pTyr sites at different developmental stages and under varying growth conditions [11], establishing the kinase/phosphatase-substrate relationship in planta via novel experimental approaches, such as the Shokat chemical genetic method [152], and CRISPR/Cas9-assisted creation of high order knockout mutants of functionally redundant DsPTKs/DUSPs in different plant species will greatly enhance our understanding of the biochemical mechanisms by which Tyr phosphorylation regulates the hormone-mediated plant growth and development.

## Figures and Tables

**Figure 1 ijms-23-06603-f001:**
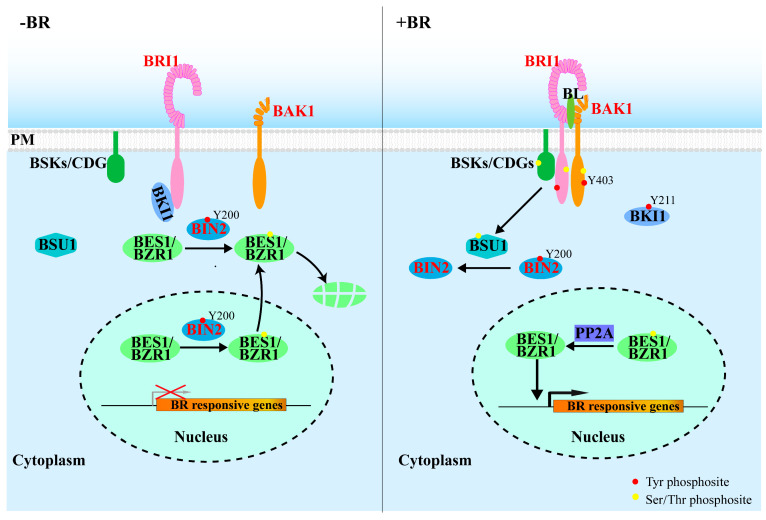
Involvement of protein Tyr phosphorylation in BR signaling. When BRs are absent (**left**), BRI1 is kept inactive at the PM (plasma membrane) by its autoinhibitory C-terminus and BKI1 association. BIN2 is constitutively active with autophosphorylated Tyr200 and phosphorylates and inhibits two homologous transcription factors: BES1/BZR1, promoting their cytosolic retention and degradation. Upon perception of BR (**right**), Tyr211 phosphorylation at BKI1 promotes its dissociation from BRI1 and moves into the cytosol, thus enabling heterodimerization and transphoshorylation of BRI1 and BAK1. The fully activated BRI1 initiates a series of Ser/Thr and Tyr phosphorylation events (autophosphorylated Tyr at both BRI1 and BAK1), resulting in phosphorylation and activation of BSU1. BUS1 subsequently inactivates BIN2 through dephosphorylation of pTyr200 at BIN2, relieving the inhibitory effect of BIN2 on BES1 and BZR1 that regulate the expression of thousands of BR-responsive genes. Name of kinases with dual-specificity are colored in red.

**Figure 2 ijms-23-06603-f002:**
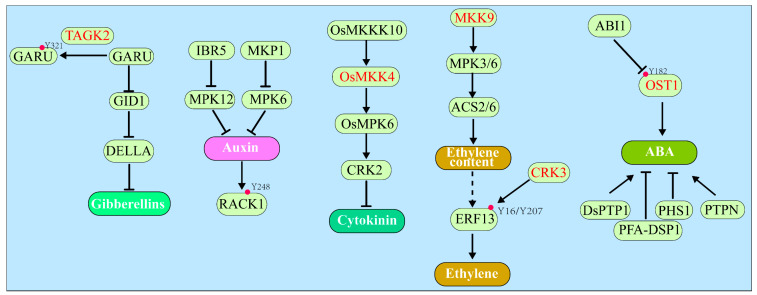
Overview of other hormones pathways regulated by the Tyr phosphorylation. In the gibberellins signaling, TAGK2 phosphorylates the GARU at Y321 and interferes with the interaction between GARU and GID1, thereby suppressing the degradation of GID1. In the ethylene signaling, CRK3 phosphorylated the Y16/Y207 of ERF13. As for the ABA signaling, OST1 can be autophosphorylated at Y182 to be fully activated, which could be dephosphorylated by ABI1. Moreover, several PTPs and MAPK cascades are implicated in the pathways of many hormones such as auxin, cytokinin, ethylene, and ABA. The → and ⊣ signs indicate the positive and negative regulatory roles between indicated proteins and the listed hormones pathways, respectively. The red circles indicate the phosphorylated Tyr residues. Names of kinases with dual-specificity are colored in red.

**Table 1 ijms-23-06603-t001:** Tyr-phosphorylated proteins in phytohormone signaling.

Hormones	Protein	Residue	Auto/Trans- Phosphorylation	Kinase	MS *	Site and p Tyr -Specific Antibody	Site-Directed Mutagenesis	Effects	Reference
BR	BRI1	Y831	Auto	BRI1	No	Yes	Yes	Reduced kinase activity	[12]
		Y956	Auto	BRI1	No	Yes	Yes	Abolished kinase activity	[12]
		Y1072	Auto	BRI1	No	Yes	Yes	Abolished kinase activity	[12]
	BKI1	Y211	Trans	BRI1	No	No	Yes	Interaction with BRI1	[14]
	BAK1	Y403	Auto	BAK1	No	Yes	Yes	Immunity response	[13]
		Y610	Auto	BAK1	No	Yes	Yes	Unclear	[61]
	BIN2	Y200	Auto	BIN2	Yes	Yes	Yes	Reduced kinase activity	[15]
GA	GARU	Y321	Trans	TAGK2	No	No	Yes	Interaction with GID1	[16]
Auxin	RACK1A	Y248	Trans	Unknown	No	No	Yes	Homo- dimerization	[65]
Cytokinin	CKX2	Y213	Trans	Unknown	Yes	No	No	Unclear	[66]
Ethylene	ERF13	Y16/207	Trans	CRK3	No	No	Yes	Unclear	[29]
ABA	OST1	Y163	Auto and Trans	OST1 and BAK1	Yes	No	Yes	Reduced kinase activity	[67]
		Y182	Auto	OST1	Yes	Yes	Yes	Reduced kinase activity	[67]

* MS, mass spectrometry.

## Data Availability

Not applicable.

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
