# Peer review of "Importance of Tyrosine Phosphorylation in Hormone-Regulated Plant Growth and Development"

_ijms, 2022, doi:10.3390/ijms23126603_

Round 1

Reviewer 1 Report

This excellent piece of work by Song et al. is very a comprehensive review on tyrosine phosphorylation in plants, with a major focus on its regulatory roles in functioning of six phytohormones. Compared to the extensive knowledge gained from the animal models, pertinent research in plant science is lagging behind. Thus, this review is very timely to provide an inventory of all information about tyrosine phosphorylation with respect to hormonal regulation. This manuscript is almost ready for publication. Minor comments are provided below for the authors’ considerations:

Fig. 1, the expression of BR-responsive genes takes place in the nucleus, not cytoplasm; on the left side (no BR), a symbol (e.g. a cross, stop sign, etc) can be used besides the gray arrow to clearly indicate that the BES1/BZR1 transcription factors are not driving the expression of BR-responsive genes.

Table 1, please consider to add another row for “ethylene” so the table completely covers all six hormones mentioned in the review; please also consider to re-order the rows, following the same flow within the text: BR, GA, auxin, cytokinin, ethylene, ABA.

L15, use the abbreviated “Tyr” for tyrosine

L34, any reference for the ratios of 86:12:2%?

L179, “was” -> “were”

L190, “inactivate” -> “inactivates”

L191, “regulate” -> “regulates”

L203-205, When I first read this sentence, I was puzzled how the residues can be phosphorylated through mutagenesis and antibodies? Please consider to rephrase, e.g. “studies using site-directed mutagenesis and phosphorylation site-specific antibodies suggest that … may be phosphorylated”.

L537 and L539, “one” -> “on”

Author Response

1) Fig. 1, the expression of BR-responsive genes takes place in the nucleus, not cytoplasm; on the left side (no BR), a symbol (e.g. a cross, stop sign, etc) can be used besides the gray arrow to clearly indicate that the BES1/BZR1 transcription factors are not driving the expression of BR responsive genes.

Response: Thanks for your comment, and we modified Fig. 1 based on your suggestions.

2) Table 1, please consider to add another row for “ethylene” so the table completely covers all six hormones mentioned in the review; please also consider to re-order the rows, following the same flow within the text: BR, GA, auxin, cytokinin, ethylene, ABA.

Response: Following your suggestion, we have re-ordered the rows of the table, following the same flow as that in the text. We also added “ethylene” into the table with ERF13 and its phosphorylating kinase CRK3.

3) L15, use the abbreviated “Tyr” for tyrosine

L34, any reference for the ratios of 86:12:2%?

L179, “was” -> “were”

L190, “inactivate” -> “inactivates”

L191, “regulate” -> “regulates”

L203-205, When I first read this sentence, I was puzzled how the residues can be

phosphorylated through mutagenesis and antibodies? Please consider to rephrase, e.g. “studies

using site-directed mutagenesis and phosphorylation site-specific antibodies suggest that …

may be phosphorylated”.

L537 and L539, “one” -> “on”

Response: Thanks for spotting these typos and making very helpful suggestions. We have corrected these mistakes, added the reference in L34, and reorganized the sentences in L203-205.

Reviewer 2 Report

The manuscript entitled "Importance of tyrosine phosphorylation in hormone-regulated plant growth and development” by Song et al. is a comprehensive review summarizing studies reporting Tyr phosphorylation in biosynthesis and signalling pathways of plant hormones required for growth and development. Authors also discuss on protein kinases and phosphatases specific to Tyr residue or displaying dual specificity towards Ser/Thr and Tyr residues.  The manuscript is well written and structured, adding valuable contribution to the understanding on the role of regulatory functions of Tyr phosphorylation in a wide range of plant growth and developmental processes. It deserves to be published in IJMS after the authors address the minor comments below.

My only concern is about the graphic aspect of the work, which I think is rather poor. I find that the only figure in the manuscript should be improved. For instance, increasing the size of the letters and ovals; improving the resolution of the pictures; indicating that the BR responsive gene is in the nucleus and not in the cytoplasm. The abbreviation PM should be explained in the figure caption as well as MS in the caption of table 1.

Moreover other figures should be proposed to help the reader to better understand implication of Tyr phosphorylation in phytohormone signalling pathways others than BR as well as the functioning of the hormone signalling pathways themselves. I strongly recommend this for auxin, ethylene and ABA.

Author Response

Thank you very much for the very positive general comment about our manuscript and helpful suggestions on figures. Here is our point-to-point responses to your suggestions>

1) My only concern is about the graphic aspect of the work, which I think is rather poor. I find that the only figure in the manuscript should be improved. For instance, increasing the size of the letters and ovals; improving the resolution of the pictures; indicating that the BR responsive gene is in the nucleus and not in the cytoplasm. The abbreviation PM should be explained in the figure caption as well as MS in the caption of table 1.

Response: Thanks for your comments. We have modified the Fig. 1 based on your suggestions. We also defined the abbreviation for PM in the figure legend and MS in the footnote of the Table.

2) Moreover other figures should be proposed to help the reader to better understand implication of Tyr phosphorylation in phytohormone signalling pathways others than BR as well as the functioning of the hormone signalling pathways themselves. I strongly recommend this for auxin, ethylene and ABA.

Response: In response to your suggestion, we added another figure (Figure 2) to illustrate the roles of the Tyr phosphorylation in regulating biosynthesis/signaling of auxin, cytokinin, ET and ABA.